biochemistry/materials science/medicinal chemistry

berberine hydrochloride, hydrogel beads, carboxymethylstarch, palygorskite, sodium alginate

**Author for correspondence:**
Jiande Gao
e-mail: 329315749@qq.com

This article has been edited by the Royal Society of Chemistry, including the commissioning, peer review process and editorial aspects up to the point of acceptance.

# Preparation and application of pH-responsive composite hydrogel beads as potential delivery carrier candidates for controlled release of berberine hydrochloride

Jiande Gao[1], Dongying Fan[2], Ping Song[1], Shudan Zhang[1] and Xiong Liu[1]

[1]College of Pharmacy, Gansu University of Traditional Chinese Medicine, Lanzhou 730000, People's Republic of China
[2]Gansu Provincial Hospital of TCM, Gansu University of Traditional Chinese Medicine, Lanzhou, People's Republic of China

JG, 0000-0002-0913-1643

For improving the effective concentration of berberine hydrochloride (BH) in the gastrointestinal tract, a series of pH-responsive hydrogel beads were prepared based on carboxymethylstarch-g-poly (acrylic acid)/palygorskite/starch/ sodium alginate (CMS-g-PAA/PGS/ST/SA) in the present work. The developed hydrogel beads were characterized by Fourier transform infrared spectroscopy (FTIR), X-ray diffraction (XRD) and thermogravimetric analysis (TG). Effect of palygorskite (PGS) content on the swelling properties of hydrogel beads and BH cumulative release were discussed. The pH responsiveness of hydrogel beads was also investigated in different media. Results illustrated that swelling of hydrogel beads and BH cumulative release from hydrogel beads were obviously affected by PGS content. The swelling ratio and BH cumulative release of composite hydrogel beads remarkably slowed down with PGS content increasing in the range from 10 to 40 wt%. The composite hydrogel beads were pH-responsive. At pH 7.4, the swelling ratio and BH cumulative release from composite hydrogel beads were the fastest among the dissolution media of pH 1.2, pH 6.8 and pH 7.4. The BH cumulative release from hydrogel beads was related to the swelling and relaxation of composite hydrogel beads and could be fitted better by the Higuchi model. The obtained composite hydrogel beads could be potentially used for the development of BH pharmaceutical dosage forms.

# 1. Introduction

Berberine hydrochloride (BH), an isoquinoline alkaloid, can be extracted from a variety of Chinese medicines such as *Hydrastis canadensis*, *Berberis aristata*, *Phellodendron amurense* and *Tinospora cordifolia* [1]. Traditionally, BH is used as a non-prescription drug to treat gastroenteritis, dysentery and abdominal pain for many years in China [2,3]. In recent years, a multitude of biological effects of BH, including anti-inflammatory [4], anti-tumour [5] and anti-hyperglycaemia [6], have also been demonstrated by numerous researches. Moreover, BH could be proven to exert the effects on mycotic infection, heart and cardiovascular diseases, and diabetic renopathy [7]. These novel bioactivities have evoked a strong desire on the potential use of BH. However, the further clinical application of BH is seriously limited due to its poor water-solubility and high polarity as an alkaloid [8,9], which resulted in low gastrointestinal absorption and bioavailability after oral administration [10]. To effectively increase the concentration of BH in the gastrointestinal tract and absorption, a new strategy for improving BH-release involves the utilization of drug carriers.

Drug carriers are an essential part for preparing new medicines in the field of pharmaceutics, which are receiving increasing attention owing to the advantage of responding to the pH value of the external environment changes, improving patient compliance, increasing drug-residence time and ensuring drug released at the desired site [11–14]. For example, Youssef *et al.* [15] prepared nanostructured lipid carriers loaded in an *in situ* gel system, which was able to prolong the residence time on the ocular surface after topical administration. Massoumi's group obtained novel pH-responsive PEGylated hollow nanocapsules, which had excellent potential for cancer chemotherapy [16]. The drug carriers are mainly from natural or synthetic polymeric hydrogels. During the past few decades, lots of natural or synthetic hydrogels such as xylan [17], chitosan [18], alginate [19], κ-carrageenan [20] and poly(*N*-isopropylacrylamide) [21], have great potential as drug carriers. However, using single polymer hydrogel as drug carrier could endow hydrogel with worse drug loading ability, biodegradability, mechanical property and water-solubility, causing the instability of drug release [22,23]. Currently, polymer/clay composite hydrogels as drug carrier exhibited encouraging results due to the synergistic effects of hydrogels and clay in biomedical and pharmaceutical applications [24,25]. For the polymer/clay composite hydrogels, literature precedents suggest that the introduction of clays could not only regulate the swelling ratio, increase the mechanical properties of the polymers but also give composite hydrogels better drug loading capacity and enhance long-term stable release [26–29]. Palygorskite (PGS), as a silicate clay with layered chain structure, has been widely used in the biological medicine industry because of its unique physical and chemical properties and low toxicity [30,31]. For example, Yahia *et al.* [32] report the comprehensive performance of palygorskite/chitosan beads could be greatly enhanced due to the synergistic effects of the palygorskite and chitosan.

Carboxymethyl starch (CMS) attracts increasing attention as an excipient in drug delivery systems due to its low price, good compaction for tablet preparation and pH-responsive properties, but pure CMS is apt to solubility causing a high burst effect of drug [33–35]. Based on the above description, the main objective of this paper is to fabricate a hydrogel bead with pH-responsive, good mechanical properties and high drug loading ability to be used as BH carrier. For this purpose, the composite materials carboxymethylstarch-g-poly (acrylic acid)/palygorskite (CMS-g-PAA/PGS) were prepared firstly. Afterwards, the resultant composite CMS-g-PAA/PGS was combined with soluble starch (ST) and sodium alginate (SA) by cross-linking SA with $Ca^{2+}$ to prepare the CMS-g-PAA/PGS/ST/SA composites hydrogels beads. The schematic of CMS-g-PAA/PGS/ST/SA composite hydrogel beads is given in scheme 1. The effects of PGS content and pH on swelling ratio, drug loading ability and drug release of the composite hydrogel beads were investigated. Meanwhile, the BH-release behaviour from the hydrogel beads was also explored for improving the effective concentration of BH in the gastrointestinal tract.

# 2. Experimental set-up

## 2.1. Materials

BH raw powder (purity > 98.2%) was purchased from Nanjing Zelang Pharmaceutical Technology Co., Ltd, China. PGS powder was obtained from Linze Colloidal Co. of Gansu province in China. CMS was purchased from Shanghai Maclean Biochemical Co., Ltd, China. Sodium alginate (SA) was from Shanghai Chemical Co. Ltd, China. Soluble starch (ST) was from Tianjin Hedong Hongyan reagent

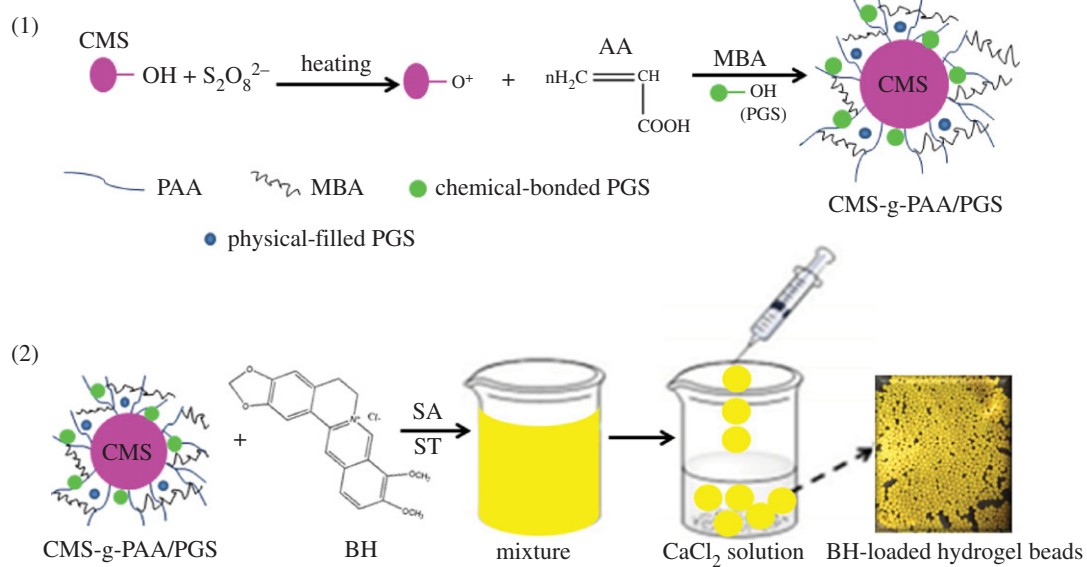

**Scheme 1.** Schematic for the preparation of composite hydrogel beads.

factory, China. *N, N′*-methylenebis acrylamide (MBA) was obtained from Drug Group Chemical Reagent Co., Ltd, China. The reagents including potassium persulfate (KPS) and acrylic acid (AA) were from Tianjin Kaixin Chemical Industrial Co., China. All solutions were prepared with distilled water.

## 2.2. Preparation of CMS-g-PAA/PGS composites

CMS-g-PAA/PGS composites with different PGS contents were prepared as our previously reported method [36,37]. CMS (2.0 g) and distilled water (25 ml) were put in a 250 ml three-necked flask equipped with a condenser, a thermometer, a nitrogen line and a mechanical stirrer. The mixture was heated to 70°C stirring for 30 min. The initiator KPS (0.01 g) was then added to the mixture solution. Under nitrogen atmosphere stirring for 15 min later, AA with a 70% neutralization degree, cross-linker MBA and PGS were added. The reaction proceeded at 70°C and was kept for 1.5 h. Subsequently, the obtained composites were washed with ethanol to remove residual reactants and homopolymer. Finally, the CMS-g-PAA/PGS composites were dried in an oven at 60°C to constant weights. The dried composites were milled and screened, and the sizes of the obtained particles were 100–120 mesh. The preparation procedure of CMS-g-PAA was similar to that of the CMS-g-PAA/PGS, except without PGS being added.

## 2.3. Preparation of CMS-g-PAA/PGS/ST/SA composite hydrogel beads

The CMS-g-PAA/PGS/ST/SA hydrogel beads were prepared as follows. Firstly, BH (0.10 g) and distilled water (100 ml) were put in a 500 ml beaker equipped with a mechanical stirrer and dissolved. CMS-g-PAA/PGS (1.0 g) microparticles were then added into the mixture solution and stirred for 3 h. After that, SA (2.0 g) and ST (5.0 g) were added and stirred for 4 h at 1000 r.p.m. Subsequently, using a 1 ml syringe the slurry was dropped into a 5 wt% CaCl₂ solution to form CMS-g-PAA/PGS/ST/SA composite hydrogel beads (SA could be cross-linked with Ca²⁺ in the CaCl₂ solution immediately). The obtained hydrogel bead products were screened and washed with distilled water several times to remove unreacted the CaCl₂ on surface, and then dried at 60°C in an oven to obtain the final product.

## 2.4. Evaluation of properties

### 2.4.1. Evaluation of BH loading

The BH-loaded CMS-g-PAA/PGS/ST/SA hydrogel beads were soaked in pH 6.8 phosphate buffer solution (PBS, 10 ml) for 12 h. Afterwards, the swollen beads were crushed and transferred into a

beaker. Subsequently, the crushed hydrogel beads were soaked in fresh PBS (20 ml) again and sonicated for 30 min to extract BH from hydrogel beads. The BH solution was centrifuged at 5000 r.p.m. for 20 min to remove the polymeric debris. UV spectrophotometer was used to analyse the BH content. The drug loading (%) was calculated using the following equation:

$$\text{drug loading (\%)} = \left(\frac{W_0}{W_1}\right) \times 100, \tag{2.1}$$

where $W_0$ represents the weight of BH in hydrogel beads, $W_1$ indicates the weights of BH-loaded composite hydrogel beads.

### 2.4.2. Evaluation of swelling properties

BH-loaded CMS-g-PAA/PGS/ST/SA hydrogel beads swelling properties were studied in pH 6.8 PBS. The 0.20 g of hydrogel beads were put into the baskets of intelligent disintegration instrument (ZRS-1C, Tianjing University Precision Instrument Factory, China) at $37 \pm 0.5°C$. At the set time intervals, the hydrogel beads were taken out from the swelling medium of pH 6.8 and weighed after removing residual liquid on the surface of hydrogel beads. The swelling ratio is calculated using the equation given below.

$$\text{swelling ratio (\%)} = \left[\frac{M_t - M_0}{M_0}\right] \times 100, \tag{2.2}$$

where $M_0$ and $M_t$ are the weight of hydrogel beads before and after soaking in the swelling medium of pH 6.8 at time $t$, respectively. The swelling ratio under various pH conditions was tested by the same procedure. The various buffer solutions were made by combining $NaH_2PO_4$, $Na_2HPO_4$, HCl and NaOH solutions. And pH meter (PHS-3E) was used to determine pH values. All swelling experiments were carried out thrice carefully under the same conditions and the average values were reported.

### 2.4.3. Evaluation of *in vitro* release of BH

*In vitro* release study of BH from BH-loaded CMS-g-PAA/PGS/ST/SA hydrogel beads was carried out as follows: 0.20 g of dried BH-loaded CMS-g-PAA/PGS/SA hydrogel beads were placed in 100 ml of the release medium, and incubated at $37 \pm 0.5°C$ under 100 r.p.m. The release medium (pH 1.2, pH 6.8 or pH 7.4) was made by combining HCl, $KH_2PO_4$ and NaOH solutions properly referring to the Chinese Pharmacopoeia 2015. At predetermined time intervals, 5 ml of the release medium was withdrawn and replaced by an equal amount of fresh release medium to keep a constant volume. The BH concentration in the release medium was assayed by UV spectrophotometer. For release medium pH 1.2, the BH concentration was assayed at 343 nm, and for release medium pH 6.8 and pH 7.4, the BH concentration was assayed at 224 nm. The BH cumulative release per cent was obtained using equation (2.3). All cumulative release results were done in triplicate.

$$\text{drug release (\%)} = \left(\frac{M_t}{M}\right) \times 100, \tag{2.3}$$

where $M$ and $M_t$ represent the BH initial amount and cumulative release amount of at time $t$, respectively. All release experiments were carried out thrice carefully under the same conditions and the average values were reported.

### 2.4.4. Analysis of *in vitro* release kinetics

The data of BH release kinetics from hydrogel beads was fitted using zero-order model, Higuchi model and Korsmeyer–Peppas model.

$$\text{Zero-order model: } \frac{M_t}{M_\infty} = Kt, \tag{2.4}$$

$$\text{Higuchi model: } \frac{M_t}{M_\infty} = Kt^{1/2} \tag{2.5}$$

and
$$\text{Korsmeyer–Peppas model: } \frac{M_t}{M_\infty} = Kt^n, \tag{2.6}$$

where $M_t/M_\infty$ represents the fraction of drug released in time $t$, $n$ is the release exponent characterizing release mechanism and $K$ is a constant. If $n \leq 0.43$, it represents that BH release from hydrogel beads is

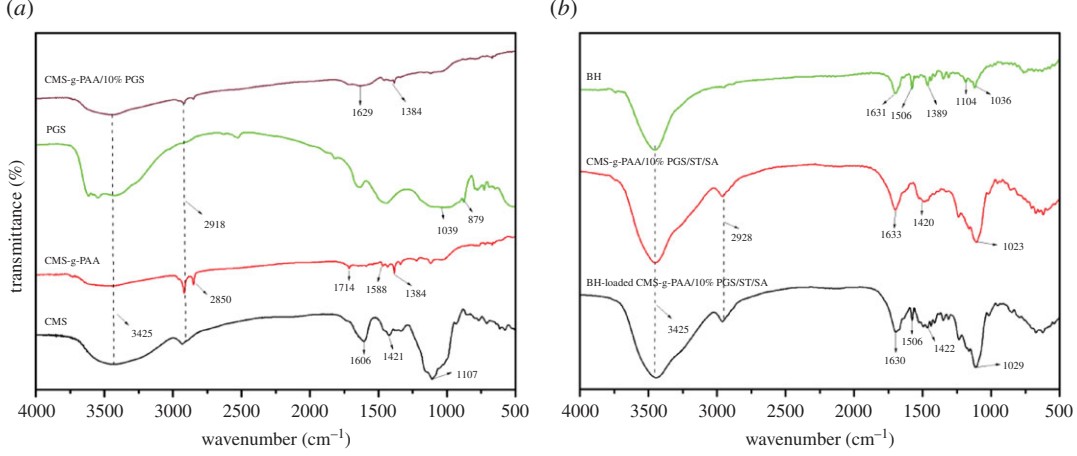

**Figure 1.** FTIR spectra of (*a*) CMS, CMS-g-PAA, PGS and CMS-g-PAA/10% PGS, and (*b*) BH, CMS-g-PAA/10% PGS/ST/SA and BH-loaded CMS-g-PAA/10% PGS/ST/SA hydrogel beads. Weight ratio of each component in the CMS-g-PAA/10% PGS/ST/SA composite hydrogel beads is that CMS-g-PAA/10% PGS : SA : ST is 1 : 2 : 5; CaCl$_2$ concentration is 5 wt%.

Fickian release. When $0.43 < n \leq 0.85$, it is defined as the non-Fickian release (both swelling-controlled release and diffusion controlled release). If $n > 0.85$, it indicates that BH release is mainly related to the relaxation (or swelling ratio) of hydrogel beads, and BH release is case II transport. It was worthwhile to note the $M_t/M_\infty$ in the Korsmeyer–Peppas model was below 60% as reported in the literature [38].

## 2.5. Characterization

Samples of FTIR spectra were taken in KBr pellets using a FTIR-FTS3000 spectrophotometer; an Ultra Plus scanning electron microscope (SEM) instrument (Carl Zeiss AG) was used to analyse micrographs; X-ray diffraction (XRD) was carried out on a Rigaku D/Max-2400 diffractometer in the $2\theta$ range of 3–80° at a scan speed of 5° min$^{-1}$; an America TA Company Instruments (TGA-Q100) was used for the thermogravimetric (TG) analysis of samples.

# 3. Results and discussion

## 3.1. FTIR spectral analysis

The FTIR spectra of CMS, CMS-g-PAA, PGS, CMS-g-PAA/10% PGS, BH, CMS-g-PAA/10% PGS/ST/SA and BH-loaded CMS-g-PAA/10% PGS/ST/SA are indicated in figure 1. As shown in figure 1*a*, CMS presented three characteristic peaks, one at 1606 cm$^{-1}$ and one at 1421 cm$^{-1}$ (attributed to vibration of –COO⁻) [39], another at 1107 cm$^{-1}$ (corresponding to the vibration of –C–O–C), which almost vanished in the spectra of CMS-g-PAA, and the peak at 3425 cm$^{-1}$ (ascribed to the vibration of –OH) weakened in the spectra of CMS-g-PAA. Meanwhile, compared with the spectra of CMS, some new peaks appeared at 2850, 1714 and 1588 cm$^{-1}$ in the spectra of CMS-g-PAA. However, the peak at 1588 cm$^{-1}$ shifted to 1626 cm$^{-1}$ in the spectra of CMS-g-PAA/10% PGS. Also, by comparison with the spectra of PGS, the peaks at 1039 cm$^{-1}$ (corresponding to the vibration of Si–O) and 879 cm$^{-1}$ (assigned to the vibration of Al–O), almost vanished in the spectra of CMS-g-PAA/10% PGS. In addition, the characteristic peaks of BH (1633, 1389, 1104 and 1036 cm$^{-1}$) disappeared in the spectra of BH-loaded CMS-g-PAA/10% PGS/ST/SA in figure 1*b*. However, the peak at 1506 cm$^{-1}$ (assigned to the vibration of –C=C [40]) still existed in the spectra of BH-loaded CMS-g-PAA/10% PGS/ST/SA. Based on the information obtained from figure 1, it could be concluded that PAA chains grafted onto the CMS, PGS also participated in the grafting copolymerization reaction, and BH was also filled in the CMS-g-PAA/10% PGS/ST/SA hydrogel beads and interacted with the composite hydrogel beads.

## 3.2. Morphological analysis

The digital photos of the BH-loaded CMS-g-PAA/10% PGS/ST/SA hydrogel beads, and the SEM images of CMS-g-PAA and CMS-g-PAA/10% PGS are shown in figure 2. As shown in figure 2*a*, swollen BH-loaded CMS-g-PAA/10% PGS/ST/SA hydrogel beads appeared as yellow spheres with

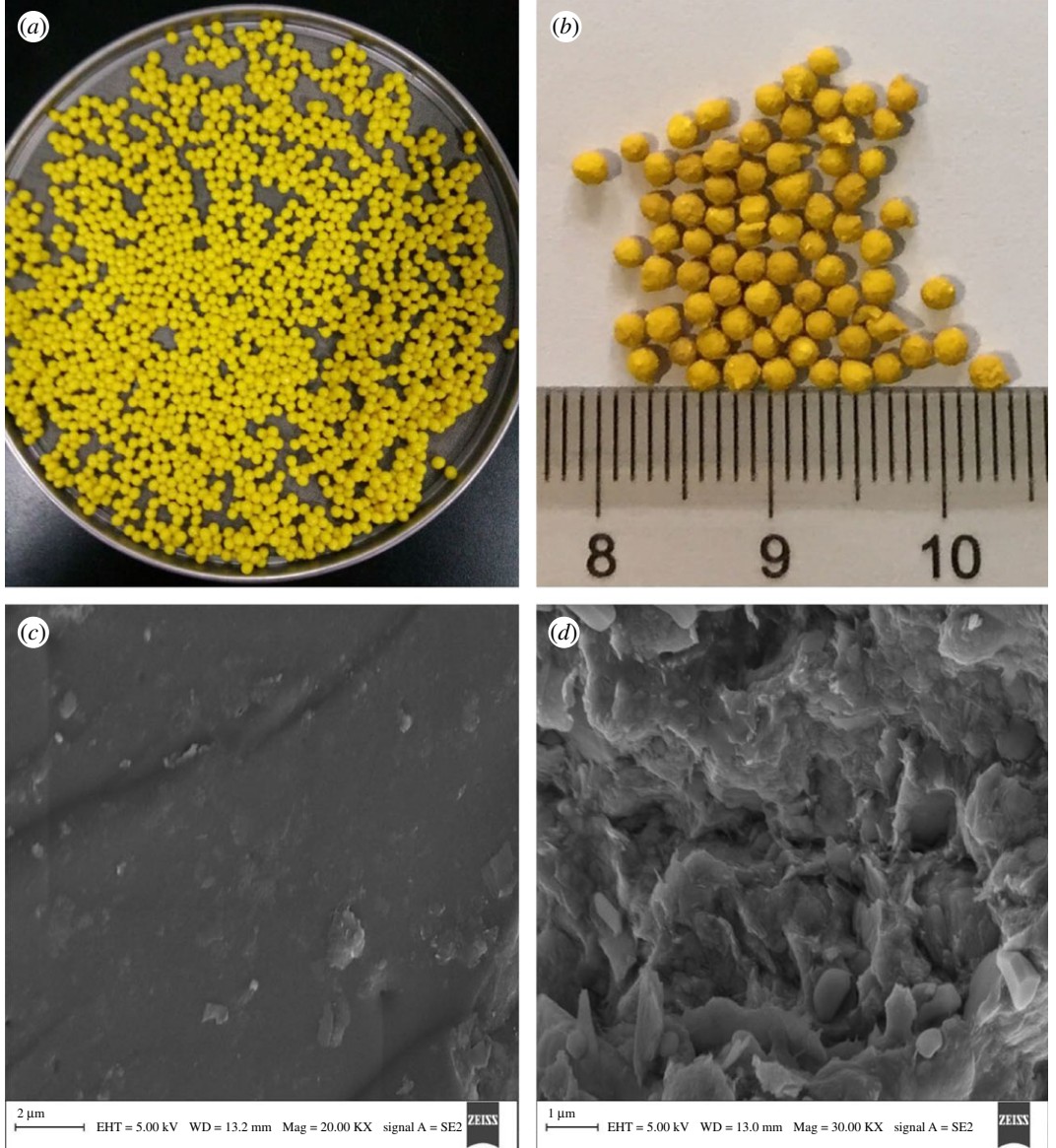

**Figure 2.** Digital photo of (*a*) BH-loaded CMS-g-PAA/10% PGS/ST/SA hydrogel beads in a swollen state and (*b*) BH-loaded CMS-g-PAA/10% PGS/ST/SA hydrogel beads in a dry state. SEM images of (*c*) CMS-g-PAA and (*d*) CMS-g-PAA/10% PGS. Weight ratio of each component in the CMS-g-PAA/10% PGS/ST/SA composite hydrogel beads is that CMS-g-PAA/10% PGS : SA : ST is 1 : 2 : 5; CaCl$_2$ concentration is 5 wt%.

smooth surfaces. After they were dried, the beads showed a rough surface and the size was approximate 2.5 mm (figure 2*b*). Additionally, CMS-g-PAA composite displayed a flat and tight surface (figure 2*c*), while CMS-g-PAA/10% PGS exhibited a relatively loose, undulant and coarse surface (figure 2*d*). Moreover, some pores could also be seen in the surface of CMS-g-PAA/10% PGS. These results indicated that the introduction of PGS could affect the specific surface area of composites, and might eventually cause the changes of swelling behaviour and drug release property [41].

## 3.3. X-ray diffraction analysis

The XRD patterns of BH, CMS-g-PAA/10% PGS/ST/SA and BH-loaded CMS-g-PAA/10% PGS/ST/SA are shown in figure 3. As could be observed, some BH crystalline structure was absent in the XRD pattern of the BH-loaded CMS-g-PAA/10% PGS/ST/SA, which indicated BH crystalline structure converted into amorphous. Moreover, the XRD profile of CMS-g-PAA/10% PGS/ST/SA showed typical characteristic peaks at $2\theta = 31.52°$ and $45.3°$. While in the XRD profile of BH-loaded CMS-g-PAA/10% PGS/ST/SA, the two characteristic peaks were weakened. It could be concluded that BH crystal

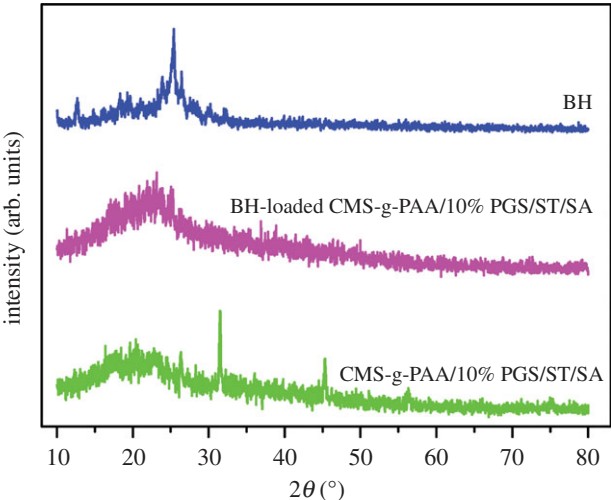

**Figure 3.** XRD patterns of BH, CMS-g-PAA/10% PGS/ST/SA and BH-loaded CMS-g-PAA/10% PGS/ST/SA composite hydrogel beads. Weight ratio of each component in the CMS-g-PAA/10% PGS/ST/SA composite hydrogel beads is that CMS-g-PAA/10% PGS : SA : ST is 1 : 2 : 5; CaCl$_2$ concentration is 5 wt%.

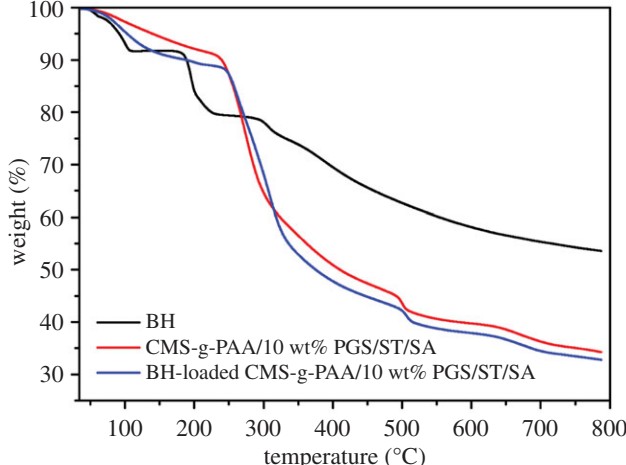

**Figure 4.** TG curves of BH, CMS-g-PAA/10% PGS/ST/SA and BH-loaded CMS-g-PAA/10% PGS/ST/SA composite hydrogel beads. Weight ratio of each component in the CMS-g-PAA/10% PGS/ST/SA composite hydrogel beads is that CMS-g-PAA/10% PGS : SA : ST is 1 : 2 : 5; CaCl$_2$ concentration is 5 wt%.

structure was changed in the BH-loaded CMS-g-PAA/10% PGS/ST/SA hydrogel beads, which might be due to the interaction between BH and CMS-g-PAA/10% PGS/ST/SA hydrogel beads.

## 3.4. Thermal stability analysis

The thermal degradation behaviour of BH, CMS-g-PAA/10% PGS/ST/SA and BH-loaded CMS-g-PAA/10% PGS/ST/SA was studied under a nitrogen atmosphere between 34 and 790°C. The results are shown in figure 4. It was evident all curves showed a decreasing trend with increasing temperature. The TG curve of BH underwent four decomposition phases. Initial weight loss of BH started at 109°C with 8.3% of weight loss, which corresponded to the loss of moisture [42]. The second step between 109 and 188°C, with 1.3% of weight loss, signified the melting temperature of the BH. While the third step showed 20.6% of weight loss at 250°C, revealing decomposition of the BH [43]. The fourth step in the 250–790°C range was ascribed to the destruction of the BH skeleton structure. However, for BH-loaded CMS-g-PAA/10% PGS/ST/SA hydrogel beads, the maximum weight loss occurred at about 265°C with 18.3% of weight loss (with 19.4% of weight loss for CMS-g-PAA/10% PGS/ST/SA at the same temperature). This meant that the thermal stability of BH in hydrogel beads suffered from minimal weight loss. The CMS-g-PAA/10% PGS/ST/SA hydrogel might improve the thermal stability through hydrogen bonding and electrostatic attraction interactions [44].

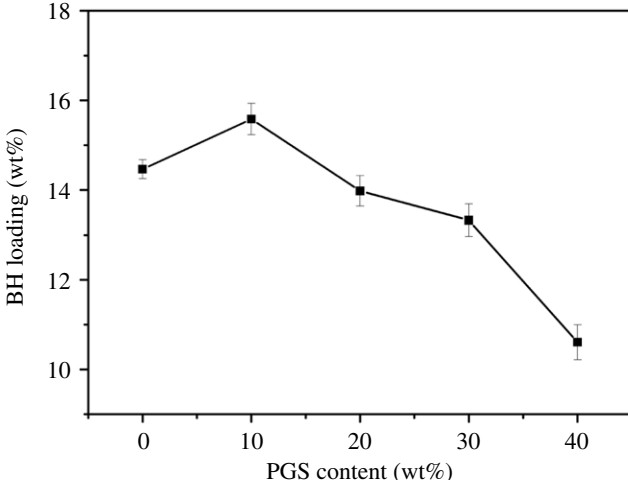

**Figure 5.** Loading of BH in the CMS-g-PAA/PGS/ST/SA composite hydrogel beads with different PGS content (wt%). Weight ratio of each component in the CMS-g-PAA/PGS/ST/SA composite hydrogel beads is that CMS-g-PAA/PGS : SA : ST is 1 : 2 : 5; CaCl$_2$ concentration is 5 wt%.

## 3.5. Effect of the PGS content on BH loading

As shown in figure 5, BH loading increased in the BH-loaded CMS-g-PAA/10% PGS/ST/SA hydrogel beads with the increase of PGS content in CMS-g-PAA/PGS composite materials from 0 to 10 wt%. The BH loading amount in the hydrogel beads decreased with further increase of PGS content in CMS-g-PAA/PGS composite materials from 10 to 40 wt%. The variation of the curve could be ascribed to the following reasons. The introduction of moderate PGS in CMS-g-PAA/PGS could relieve the entanglement of the polymeric chains, which could be beneficial to decrease the physical cross-linking degree and facilitate the entrapment of drug in CMS-g-PAA/PGS. Consequently, the higher drug loading was observed in the hydrogel beads [45]. However, the additional PGS could also react with AA and CMS [46], which led to more cross-linking points in CMS-g-PAA/PGS and restricted the entrapment of BH in CMS-g-PAA/PGS, and then caused the decrease of the BH loading.

## 3.6. Swelling properties

### 3.6.1. Effect of the PGS content on swelling

Clay content has a very important effect on the swelling of polymer/clay composites because clay could affect the network structure of polymer/clay composites [47]. Thus, the effect of PGS content on swelling performance of the CMS-g-PAA/PGS/ST/SA hydrogel beads in pH 6.8 PBS was studied. As shown in figure 6, when 10 wt% of PGS was introduced in the CMS-g-PAA/PGS composite material, the swelling ratio of the CMS-g-PAA/10 wt% PGS/ST/SA hydrogel beads reached maximum. Compared with CMS-g-PAA/ST/SA hydrogel beads, the equilibrium swelling rate of CMS-g-PAA/10 wt% PGS/ST/SA hydrogel beads significantly increased. However, when PGS content ranged from 10 to 40 wt% in the CMS-g-PAA/PGS composites, the swelling ratio of CMS-g-PAA/PGS/ST/SA hydrogel beads decreased with increasing PGS content. According to previous reports [48,49], the reason might be ascribed to the fact that moderate PGS was able to impair the hydrogen-bonding interaction and the entanglement of the polymeric chains, which improved the swelling ratio of hydrogel beads. Nevertheless, excess PGS took up a position of cross-linking points in hydrogel beads by large amounts of –OH groups on the surface of PGS. Meanwhile, the excessive PGS may enhance the cross-linking density of composite and plugging network voids and minimize the swelling ratio.

### 3.6.2. Effect of pH on swelling

Figure 7 indicates the swelling ratio variation of the CMS-g-PAA/10 wt% PGS/ST/SA hydrogel beads in different media. When pH of the media was 1.2, the swelling of the composite hydrogel beads was very small and the average swelling rate was 5.14 within 10 h. At pH 6.8, the average swelling rate was 8.06, 14.87 and 17.24 after 2, 5 and 10 h, respectively. While at pH 7.4, the average swelling rate was 8.40, 15.47

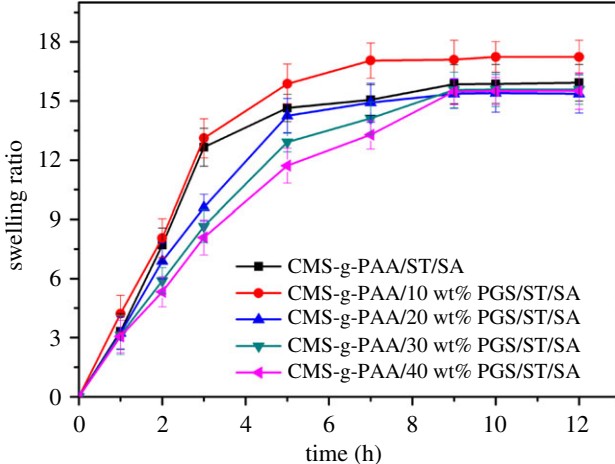

**Figure 6.** Variation of swelling ratio for CMS-g-PAA/PGS/ST/SA composite hydrogel beads with different PGS content in pH 6.8. Weight ratio of each component in the CMS-g-PAA/PGS/ST/SA composite hydrogel beads is that CMS-g-PAA/PGS : SA : ST is 1 : 2 : 5; CaCl₂ concentration is 5 wt%.

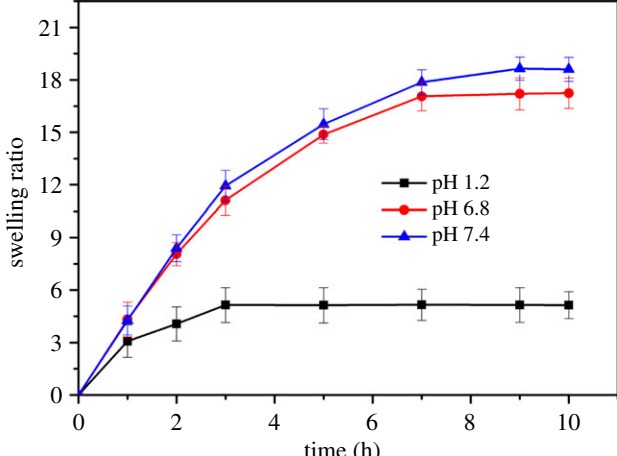

**Figure 7.** Variation of swelling ratio for CMS-g-PAA/10% PGS /ST/SA composite hydrogel beads at different pH solution. Weight ratio of each component in the CMS-g-PAA/10% PGS/ST/SA composite hydrogel beads is that CMS-g-PAA/10% PGS : SA : ST is 1 : 2 : 5; CaCl₂ concentration is 5 wt%.

and 18.61 after 2, 5 and 10 h, respectively. The transformation of swelling behaviour with pH is attributed to the fact that most of –COO⁻ groups converted to –COOH groups in the acidic solution, which strengthened the hydrogen-bonding interaction of the polymeric chains in CMS-g-PAA/10 wt% PGS/ ST/SA hydrogel beads causing the small swelling ratio. At pH 6.8, the electrostatic repulsion of –COO⁻ within the test hydrogel beads weakened the hydrogen-bonding interaction of the polymeric chains, which made the hydrogel beads swollen. With further increase of pH to 7.4, the hydrogen-bonding interaction of hydrogel beads disintegrated and the electrostatic repulsion of –COO⁻ within the test hydrogel beads strengthened, which led to the further increase of the swelling ratio. The above information indicated the hydrogel beads had a benign pH-responsive behaviour with an external pH solution, which was also similar to the previous research reports in the literature [50,51].

## 3.7. *In vitro* release of BH

### 3.7.1. Effect of the PGS content on BH release

Figure 8 shows the effect of the PGS content on the cumulative release of BH from BH-loaded CMS-g-PAA/10% PGS/ST/SA hydrogel beads in pH 6.8 PBS. It was indicated obviously that when PGS content ranged from 10 to 40 wt% in the CMS-g-PAA/PGS composites, BH release decreased from

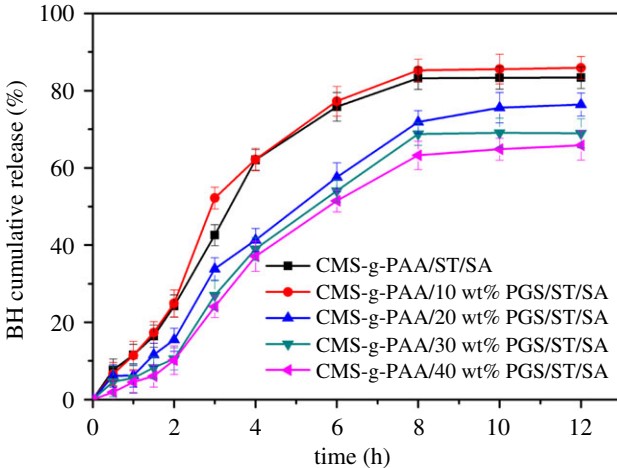

**Figure 8.** The cumulative release of BH from CMS-g-PAA/PGS/ST/SA composite hydrogel beads with different PGS content at 37 ± 0.5°C under 100 r.p.m. after 12 h. Weight ratio of each component in the CMS-g-PAA/PGS/ST/SA composite hydrogel beads is that CMS-g-PAA/PGS : SA : ST is 1 : 2 : 5; CaCl$_2$ concentration is 5 wt%.

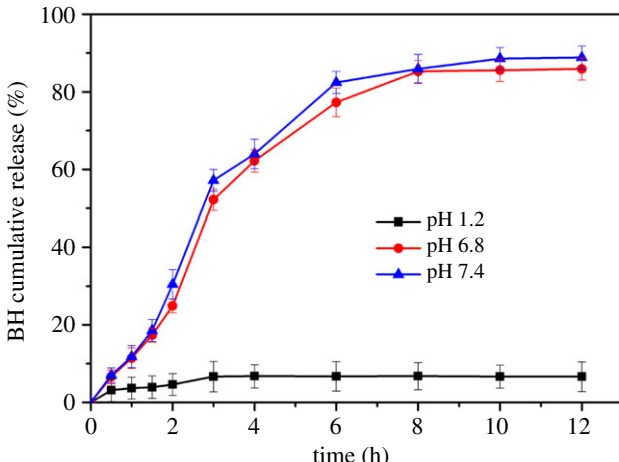

**Figure 9.** *In vitro* cumulative release of BH from CMS-g-PAA/10% PGS/ST/SA composite hydrogel beads at various pHs at 37 ± 0.5°C under 100 r.p.m. after 12 h. Weight ratio of each component in the CMS-g-PAA/10% PGS/ST/SA composite hydrogel beads is that CMS-g-PAA/10% PGS : SA : ST is 1 : 2 : 5; CaCl$_2$ concentration is 5 wt%.

composite hydrogel beads with increasing PGS content. The cumulative release ratio of BH from the BH-loaded CMS-g-PAA/10% PGS/ST/SA composite hydrogel beads was 17.36% (1.5 h) to 77.27% (6 h), and up to 85.90% (12 h). However, when PGS content increased to 40%, the cumulative release ratio of BH was 6.05% within 1.5 h, 51.43% within 6 h and 65.81% within 12 h, while for BH-loaded CMS-g-PAA/ST/SA hydrogel beads, the cumulative release ratio of BH was 16.41%, 75.82% and 83.38% after 1.5, 6 and 12 h, respectively, which was faster than that of the BH-loaded CMS-g-PAA/ 40 wt% PGS/ST/SA composite hydrogel beads. The phenomenon could be ascribed to the facts the introduction of PGS could improve the surface area of composite hydrogel beads, which enhanced the adsorption of the BH on the surface of PGS and then BH migrated from hydrogel beads through a longer path. The result led to a slower BH release. In addition, excess PGS could restrict the swelling rate of hydrogel beads and the BH release might be lower.

### 3.7.2. Effect of pH on the release of BH

The effect of pH on BH release from BH-loaded CMS-g-PAA/10% PGS/ST/SA hydrogel beads is shown in figure 9. It could be observed that for the composite hydrogel beads in the medium of pH 1.2, the BH cumulative release ratio from the BH-loaded CMS-g-PAA/10 wt% PGS/ST/SA composite hydrogel beads was 3.68% for 1 h, and 6.77% for 8 h, while in pH 6.8 PBS, the BH cumulative release ratio was

**Table 1.** Model fitting for release mechanism of hydrogel beads at different PGS content: kinetic constants ($K$), release exponent ($n$) and correlation coefficient ($R$).

| PGS (wt%) | N | Korsmeyer–Peppas model | | Higuchi model | | zero-order model | |
|---|---|---|---|---|---|---|---|
| | | $K \times 10^2$ | $R^2$ | $K \times 10^2$ | $R^2$ | $K \times 10^2$ | $R^2$ |
| 0 | 0.9451 | 0.1293 | 0.9572 | 0.3009 | 0.9266 | 0.0766 | 0.8574 |
| 10 | 1.1221 | 0.1256 | 0.9636 | 0.3093 | 0.9247 | 0.0783 | 0.8467 |
| 20 | 1.0306 | 0.0890 | 0.9298 | 0.2705 | 0.9433 | 0.0712 | 0.9332 |
| 30 | 1.1064 | 0.0686 | 0.9238 | 0.2534 | 0.9216 | 0.0670 | 0.9188 |
| 40 | 1.397 | 0.0447 | 0.9807 | 0.2423 | 0.9220 | 0.0642 | 0.9231 |

11.40% for 1 h and 85.25% for 8 h. Nevertheless, the BH cumulative release ratio quickened remarkably at pH 7.4, which was 11.79%, 85.92% and 88.86% after 1, 8 and 12 h, respectively. Obviously, at pH 1.2, the BH release from BH-loaded CMS-g-PAA/10 wt% PGS/ST/SA hydrogel beads was the slowest in the three kinds of release media. The phenomenon may be mostly owing to swelling behaviour variation in different pH media. It was difficult for the BH to release from the bead at pH 1.2 due to the shrinking of the beads. However, the swelling ratio of the beads increased at pH 6.8 or 7.4. Consequently, the increase of swelling ratio for the beads facilitated the migration of BH from the composite hydrogel beads. A similar phenomenon also was reported by Yin *et al.* [52].

### 3.7.3. Model fitting of BH release mechanism

The results of model fitting are shown in table 1. It could be found the BH release behaviour from the composite hydrogel beads fitted the Higuchi equation ($R^2 = 0.9216$–0.9433) better than the zero-order equation ($R^2 = 0.8467$–0.9332) in pH 6.8 PBS according to the respective correlation coefficients ($R$). For the relationship between BH cumulative release from hydrogel beads and the relaxation as well as erosion of hydrogel beads, the Korsmeyer–Peppas model showed all the $n$ values were between 0.9451 and 1.397, which means that BH release mechanism was mainly related to the swelling and relaxation of composite hydrogel beads.

## 4. Conclusion

In this paper, a series of new pH-responsive composite hydrogel beads were successfully developed in order to improve the effective concentration of BH in the gastrointestinal tract. The incorporation of PGS could influence the amount of BH loading in hydrogel beads. The swelling ratio of hydrogel beads and BH cumulative release decreased with the increase of PGS content in hydrogel beads. The swelling ratio of hydrogel beads and BH cumulative release from hydrogel beads at pH 7.4 were faster than that at pH 6.8. BH cumulative release from hydrogel beads was related to the swelling ratio and relaxation of composite hydrogel beads and could be described better by the Higuchi model. These obtained results showed that the incorporation of PGS in composite hydrogel beads could regulate drug load and release, which would facilitate the preparation of novel drug carriers. The composite hydrogel beads is useful for improving the concentration of BH in the gastrointestinal tract. The method of preparation for hydrogel beads was also proved to be simple.

Data accessibility. The datasets supporting this article have been uploaded as part of the electronic supplementary material.

Authors' contributions. J.G. is the first author because he established the experimental programme and drafted the manuscript. D.F. collected literature and recorded experimental data. S.Z. helped J.G. complete the manuscript modification. P.S. participated in data analysis. X.L. and J.G. conceived of, designed and coordinated the study, and helped draft the manuscript. All the authors gave their final approval for publication.

Competing interests. We declare we have no competing interests.

Funding. This work was supported by the Natural Science Foundation of Gansu Province (grant no. 17JR5RA172), Gansu Province Science and Technology Research (grant no. GYC13-05).

Acknowledgements. The author and co-workers would like to thank the Natural Science Foundation of Gansu Province and Gansu Province Science and Technology Research. They provided the authors with funding support.

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
