## [Reviewer comments · Royal Society Open Science]

Review History

RSOS-200676.R0 (Original submission)

Review form: Reviewer 1

Is the manuscript scientifically sound in its present form?

No

Are the interpretations and conclusions justified by the results?

Yes

Is the language acceptable?

Yes

Do you have any ethical concerns with this paper?

No

Have you any concerns about statistical analyses in this paper?

Yes

Recommendation?

Major revision is needed (please make suggestions in comments)

Comments to the Author(s)

In this concise manuscript, the authors reported the preparation and application of pH-responsive composite hydrogel beads as potential delivery carrier candidates for the controlled release of berberine hydrochloride. The obtained composite hydrogel beads could be potentially utilized for the development of BH7 pharmaceutical dosage forms. However, the following issues should be addressed or corrected before publication.

1. The structure of BH, polymer structure, and a scheme of overall design should be provided for easy understanding.
2. This is quite complicated materials. Adding the control of sodium alginate gel may justify such structure. Besides, what was the ratio of each component?
3. Swelling and release experiments should be conducted at least three times
4. Is the swelling of the hydrogel at a range of 3-15%? This is extremely low! It essentially did not swell, just absorbed some moisture.
5. All figure captions should contain more information.
6. What were the BH loading efficiencies?
7. In figure 1a, peaks were quite small, especially CMS-g-PAA. The quality should be improved.
8. In figure 2, captions were wrong. There are four images.
9. In figure 4, please add PGS curve
10. In figure 5, what would be loading content if PGS content were deducted? How did the author calculate the PGS content?
11. What is the solubility of BH at pH 7.4? Do the authors observe any precipitations during release?
12. CMS-g-PAA should be characterized by proton NMR to determine the structure
13. The full name of MBA should be provided.
14. In the release experiment, what is the concentration of total ions? Is PBS the buffer used.

Review form: Reviewer 2

Is the manuscript scientifically sound in its present form?

Yes

Are the interpretations and conclusions justified by the results?

Yes

Is the language acceptable?

Yes

Do you have any ethical concerns with this paper?

No

Have you any concerns about statistical analyses in this paper?

Yes

Recommendation?

Accept with minor revision (please list in comments)

Comments to the Author(s)

1. What are the error bars in Figure 7, 8, 9?
2. Do you have statistical analysis of Figure 2 and Figure 4?
3. I don't see much difference in Figure 6. Could you explain more on Figure 6?

4. There are several issues with the grammar and English. Please revise the manuscript thoroughly.
5. A large body of latest literature on biopolymer-based materials for biomedical applications is missing and must be discussed and referenced in their work. This is important so that readers are aware of the current literature. They are:
 - 1) Treatment of Neurodegenerative Disorders through the Blood-brain Barrier using Nanocarriers. *Nanoscale* 2018, 10(36): 16962-16983.
 - 2) Alternative Foaming Agents for Topical Treatment of Ulcerative Colitis. *Journal of Biomedical Materials Research Part A* 2018, 106(5): 1448-1456.
 - 3) Non-genetic Engineering of Cells for Drug Delivery and Cell-based Therapy. *Advanced Drug Delivery Reviews* 2015, 91(1): 125-140.
 - 4) Intestinal Organoids Containing PLGA Nanoparticles for the Treatment of Inflammatory Bowel Diseases. *Journal of Biomedical Materials Research Part A* 2018, 106(4): 876-886.
 - 5) A Bio-responsive 6-mercaptapurine/Doxorubicin Based "Click Chemistry" Polymeric Prodrug for Cancer Therapy. *Materials Science and Engineering: C* 2020, 108(1): 110461- 110473.
 - 6) Synthesis and In Vitro Characterization of Carboxymethyl Chitosan-CBA-Doxorubicin Conjugate Nanoparticles as pH-Sensitive Drug Delivery Systems. *Journal of Biomedical Nanotechnology* 2017, 13(9): 1097-1105.
 - 7) Self-Assembled Hyaluronic Acid Nanoparticles for pH-Sensitive Release of Doxorubicin: Synthesis and In Vitro Characterization. *Journal of Biomedical Nanotechnology* 2017, 13(9), 1058-1068.
 - 8) A pH-Sensitive Nanosystem Based on Carboxymethyl Chitosan for Tumor-Targeted Delivery of Daunorubicin. *Journal of Biomedical Nanotechnology* 2016, 12(8): 1688-1698.

Decision letter (RSOS-200676.R0)

Dear Dr Gao:

Title: Preparation and application of pH-responsive hydrogel beads as potential delivery carrier candidates for controlled release of berberine hydrochloride
Manuscript ID: RSOS-200676

The editor assigned to your manuscript has now received comments from reviewers. We would like you to revise your paper in accordance with the referee and Subject Editor suggestions which can be found below (not including confidential reports to the Editor). Please note this decision does not guarantee eventual acceptance.

Please submit your revised paper before 18-Jul-2020. Please note that the revision deadline will expire at 00.00am on this date. If we do not hear from you within this time then it will be assumed that the paper has been withdrawn. In exceptional circumstances, extensions may be possible if agreed with the Editorial Office in advance. We do not allow multiple rounds of revision so we urge you to make every effort to fully address all of the comments at this stage. If deemed necessary by the Editors, your manuscript will be sent back to one or more of the original reviewers for assessment. If the original reviewers are not available we may invite new reviewers.

On behalf of the Subject Editor Professor Anthony Stace and the Associate Editor Dr Andrew Harned.

RSC Associate Editor:

Comments to the Author:

The reviewers have expressed enthusiasm for the work as a whole, but have noted several comments/concerns that need to be addressed in a revised manuscript. Many of these, particularly those of Reviewer 1, address valid scientific concerns.

Given, the present global circumstances, I ask the authors to do their best at addressing these concerns. There may be some concerns that require additional experimentation. If this is not possible, please justify why we should consider a revised manuscript without these experiments.

RSC Subject Editor:

Comments to the Author:

(There are no comments.)

Reviewers' Comments to Author:

Reviewer: 1

Comments to the Author(s)

In this concise manuscript, the authors reported the preparation and application of pH-responsive composite hydrogel beads as potential delivery carrier candidates for the controlled release of berberine hydrochloride. The obtained composite hydrogel beads could be potentially utilized for the development of BH7 pharmaceutical dosage forms. However, the following issues should be addressed or corrected before publication.

1. The structure of BH, polymer structure, and a scheme of overall design should be provided for easy understanding.
2. This is quite complicated materials. Adding the control of sodium alginate gel may justify such structure. Besides, what was the ratio of each component?
3. Swelling and release experiments should be conducted at least three times
4. Is the swelling of the hydrogel at a range of 3-15%? This is extremely low! It essentially did not swell, just absorbed some moisture.
5. All figure captions should contain more information.
6. What were the BH loading efficiencies?
7. In figure 1a, peaks were quite small, especially CMS-g-PAA. The quality should be improved.
8. In figure 2, captions were wrong. There are four images.
9. In figure 4, please add PGS curve
10. In figure 5, what would be loading content if PGS content were deducted? How did the author calculate the PGS content?
11. What is the solubility of BH at pH 7.4? Do the authors observe any precipitations during release?
12. CMS-g-PAA should be characterized by proton NMR to determine the structure
13. The full name of MBA should be provided.
14. In the release experiment, what is the concentration of total ions? Is PBS the buffer used.

Reviewer: 2

Comments to the Author(s)

1. What are the error bars in Figure 7, 8, 9?
2. Do you have statistical analysis of Figure 2 and Figure 4?
3. I don't see much difference in Figure 6. Could you explain more on Figure 6?
4. There are several issues with the grammar and English. Please revise the manuscript thoroughly.
5. A large body of latest literature on biopolymer-based materials for biomedical applications is missing and must be discussed and referenced in their work. This is important so that readers are aware of the current literature. They are:
 - 1) Treatment of Neurodegenerative Disorders through the Blood-brain Barrier using Nanocarriers. *Nanoscale* 2018, 10(36): 16962-16983.
 - 2) Alternative Foaming Agents for Topical Treatment of Ulcerative Colitis. *Journal of Biomedical Materials Research Part A* 2018, 106(5): 1448-1456.
 - 3) Non-genetic Engineering of Cells for Drug Delivery and Cell-based Therapy. *Advanced Drug Delivery Reviews* 2015, 91(1): 125-140.
 - 4) Intestinal Organoids Containing PLGA Nanoparticles for the Treatment of Inflammatory Bowel Diseases. *Journal of Biomedical Materials Research Part A* 2018, 106(4): 876-886.
 - 5) A Bio-responsive 6-mercaptopurine/Doxorubicin Based "Click Chemistry" Polymeric Prodrug for Cancer Therapy. *Materials Science and Engineering: C* 2020, 108(1): 110461-110473.
 - 6) Synthesis and In Vitro Characterization of Carboxymethyl Chitosan-CBA-Doxorubicin Conjugate Nanoparticles as pH-Sensitive Drug Delivery Systems. *Journal of Biomedical Nanotechnology* 2017, 13(9): 1097-1105.
 - 7) Self-Assembled Hyaluronic Acid Nanoparticles for pH-Sensitive Release of Doxorubicin: Synthesis and In Vitro Characterization. *Journal of Biomedical Nanotechnology* 2017, 13(9), 1058-1068.
 - 8) A pH-Sensitive Nanosystem Based on Carboxymethyl Chitosan for Tumor-Targeted Delivery of Daunorubicin. *Journal of Biomedical Nanotechnology* 2016, 12(8): 1688-1698.

Author's Response to Decision Letter for (RSOS-200676.R0)

See Appendix A.

RSOS-200676.R1 (Revision)

Review form: Reviewer 2

Is the manuscript scientifically sound in its present form?

Yes

Are the interpretations and conclusions justified by the results?

Yes

Is the language acceptable?

Yes

Do you have any ethical concerns with this paper?

Yes

Have you any concerns about statistical analyses in this paper?

Yes

Recommendation?

Accept as is

Comments to the Author(s)

All the previous concerns have been properly addressed. The manuscript is acceptable at current status.

Decision letter (RSOS-200676.R1)

Dear Dr Gao:

Title: Preparation and application of pH-responsive hydrogel beads as potential delivery carrier candidates for controlled release of berberine hydrochloride

Manuscript ID: RSOS-200676.R1

It is a pleasure to accept your manuscript in its current form for publication in Royal Society Open Science. The chemistry content of Royal Society Open Science is published in collaboration with the Royal Society of Chemistry.

On behalf of the Subject Editor Professor Anthony Stace and the Associate Editor Dr Andrew Harned.

RSC Associate Editor:
Comments to the Author:
The authors appear to have addressed the concerns raised in the previous review.

RSC Subject Editor:
Comments to the Author:
(There are no comments.)

Reviewer(s)' Comments to Author:
Reviewer: 2

Comments to the Author(s)
All the previous concerns have been properly addressed. The manuscript is acceptable at current status.

Appendix A

Response to the Reviewer's Comments on manuscript:

Title: "Preparation and application of pH-responsive hydrogel beads as potential delivery carrier candidates for controlled release of berberine hydrochloride". (ID: RSOS-200676).

The authors are very thankful to the reviewers for their valuable suggestions and comments to improve the quality of our manuscript. The necessary modifications and corrections have carefully been made as reviewer's suggestions and the revised parts have been highlighted in **blue** color text in the revised manuscript.

Reviewer(s)' Comments to Author:

Reviewer: 1

Comments:

1. The structure of BH, polymer structure, and a scheme of overall design should be provided for easy understanding.

Our response: Thank you very much for reviewers' comment. we have provided the structure of BH, polymer structure, and a scheme of overall design for easy understanding, as shown in the follow figure and in **Scheme 1** in manuscript.

2. This is quite complicated materials. Adding the control of sodium alginate gel may justify such structure. Besides, what was the ratio of each component?

Our response: Thank you very much for reviewers' positive comments. Currently, polymer/clay composite hydrogels as drug carrier exhibited encouraging results due to the synergistic effects of hydrogels and clay in biomedical and pharmaceutical applications. Literatures of the polymer/clay composite hydrogels report that the introduction of clays could regulate the swelling ratio, increase the mechanical properties of the polymers, endow composite hydrogels better drug loading capacity and long-term stable release. However, our group found that drugs in the single polymer hydrogels beads would overflow and the instability of drug-release. In order to prevent the spillover and effectively control the release of drugs, our group designed this approach in manuscript. Besides, the weight ratio of each component in the composite hydrogel beads is that CMS-g-PAA/PGS : SA :ST is 1:2:5.

Literature of the polymer/clay composite hydrogels is as follows:

[24 in manuscript] Wu J, Ding SJ, Chen J. 2014. Preparation and drug release properties of chitosan/organomodified palygorskite microspheres. *Int. J. Biol. Macromol.* 68: 107-112.

[25 in manuscript] Yang HX, Wang WB, Zhang JP, Wang AQ. 2013. Preparation, Characterization, and Drug-Release Behaviors of a pH-Sensitive Composite Hydrogel Bead Based on Guar Gum, Attapulgit, and Sodium Alginate. *Int. J. Polym. Mater. Polym. Biomater.* 62: 369-376.

[29 in manuscript] Wang Q, Xie XL, Zhang XW. 2010. Preparation and swelling properties of pH-sensitive composite hydrogel beads based on chitosan-g-poly (acrylic acid)/vermiculite and sodium alginate for diclofenac controlled release. *Int. J. Biol. Macromol.* 46: 356-362.

[30 in manuscript] Wang Q, Wang WB, Wu J, Wang AQ. 2012. Effect of Attapulgit Contents on Release Behaviors of a pH Sensitive Carboxymethyl Cellulose-g-Poly(acrylic acid)/Attapulgit/Sodium Alginate Composite Hydrogel Bead Containing Diclofenac. *J. Appl. Polym. Sci.* 124, 4424-4432.

3. Swelling and release experiments should be conducted at least three times

Our response: Thank you very much for reviewers' constructive suggestions. Swelling and release experiments were carried out thrice carefully under the same conditions and the average values were reported.

4. Is the swelling of the hydrogel at a range of 3-15%? This is extremely low! It essentially did not

swell, just absorbed some moisture.

Our response: Thank you very much for reviewers' constructive suggestions. The swelling ratio of the hydrogel beads is indeed small, which is related to pH and time. When pH of the media was 1.2, the swelling of the composite hydrogel beads was very small and the average swelling rate was 5.14 within 10 h. At pH 6.8, the average swelling rate was 8.06, 14.87 and 17.24 after 2 h, 5 h, and 10 h, respectively. While at pH 7.4, the average swelling rate was 8.40, 15.47 and 18.61 after 2 h, 5 h, and 10 h, respectively. The relevant literature is as follows:

[29 in manuscript] Wang Q, Xie XL, Zhang XW. 2010. Preparation and swelling properties of pH-sensitive composite hydrogel beads based on chitosan-g-poly (acrylic acid)/vermiculite and sodium alginate for diclofenac controlled release. *Int. J. Biol. Macromol.* 46: 356-362.

[41 in manuscript] Wang Q, Zhang JP, Wang AQ. 2009. Preparation and characterization of a novel pH-sensitive chitosan-g-poly (acrylic acid)/attapulgitite/sodium alginate composite hydrogel bead for controlled release of diclofenac sodium. *Carbohydr. Polym.* 78: 731-737.

5. All figure captions should contain more information.

Our response: Thank you very much for reviewers' constructive suggestions. According to the reviewers' suggestions, we have provided more information for some figure captions.

6. What were the BH loading efficiencies?

Our response: Thank you very much for reviewers' constructive suggestions. In fact, what we want to express is the amount of the BH in CMS-g-PAA/PGS/ST/SA hydrogel beads. Drug loading (%) = $(W_0 / W_1) \times 100$; where W_0 represent the weight of BH in hydrogel beads, W_1 indicates the weights of BH-loaded composite hydrogel beads.

7. In figure 1a, peaks were quite small, especially CMS-g-PAA. The quality should be improved.

Our response: Thank you very much for reviewers' positive comments. We think that the comment is very constructive. As far as we know, the graft copolymerization of CMS-g-PAA could be proved by performance test, such as water absorbency. Similar reactions have been done before. Related literatures as follows:

[36 in manuscript] Gao JD, Yang Q, Ran FT, Ma GF, Lei ZQ. 2016. Preparation and properties of novel eco-friendly superabsorbent composites based on raw wheat bran and clays. *Appl. Clay. Sci.*

132: 739-747.

8. In figure 2, captions were wrong. There are four images.

Our response: Thank you very much for reviewers' positive comments. According to the reviewers' suggestions, we have revised the captions In figure 2.

9. In figure 4, please add PGS curve

Our response: Thank you very much for reviewers' positive comments. We think that the comment is very constructive for our later work. As far as figure 4 is concerned, we attempted to show the thermal stability of BH, which was improved after adding the CMS-g-PAA/10 wt% PGS/ST/SA composite hydrogel beads, and the composite hydrogel beads might improve the thermal stability through hydrogen bonding and electrostatic attraction interactions.

10. In figure 5, what would be loading content if PGS content were deducted? How did the author calculate the PGS content?

Our response: Thank you very much for reviewers' positive comments. If PGS content were deducted, the BH loading (%) would be $(W_0 / W_1) \times 100$; W_0 represents the weight of BH in CMS-g-PAA/ST/SA composite hydrogel beads, W_1 indicates the weights of BH-loaded CMS-g-PAA/ST/SA composite hydrogel beads. The PGS content is for CMS-g-PAA/PGS composite material. For example, 10 wt% PGS in the CMS-g-PAA/10 wt% PGS is the total of CMS、AA、NaOH、KPS、MBA and PGS.

11. What is the solubility of BH at pH 7.4? Do the authors observe any precipitations during release?

Our response: Thank you very much for reviewers' positive comments. The solubility of BH in water is about 1:500. In the manuscript, 0.20 g of dried BH-loaded CMS-g-PAA/PGS/SA hydrogel beads were placed in 100 mL of the release medium, and incubated at 37 ± 0.5 °C under 100 rpm. The authors and co-authors could not observe precipitation during release.

12. CMS-g-PAA should be characterized by proton NMR to determine the structure

Our response: we appreciate the reviewer very much for offering us such constructive suggestions.

As far as we know, the graft copolymerization is usually proved by simple and efficient traditional characterization methods, such as fourier transform infrared spectroscopy (FTIR), X-ray diffraction (XRD) and thermogravimetric analysis (TG). In addition, we also cite the relevant discussion results

literatures in manuscript to further demonstration of our experimental results. The relevant literatures of the graft copolymerization (CMS-g-PAA) are as follows:

[1] Cheng S, Liu XM, Zhen JH, Lei ZQ. 2019. Preparation of superabsorbent resin with fast water absorption rate based on hydroxymethyl cellulose sodium and its application. *Carbohydr. Polym.* <https://doi.org/10.1016/j.carbpol.2019.115214>.

[2] Kenawy E, Seggiani M, Cinelli P, Elnaby HMM, Azaam MM. 2020. Swelling capacity of sugarcane bagasse-g-poly(acrylamide)/attapulgit superabsorbent composites and their application as slow release fertilizer. *Eur. Polym. J.* <https://doi.org/10.1016/j.eurpolymj.2020.109769>.

[3] Behrouzi M, Moghadam PN. 2018. Synthesis of a new superabsorbent copolymer based on acrylic acid grafted onto carboxymethyl tragacanth. *Carbohydr. Polym.* 202: 227-235.

13. The full name of MBA should be provided.

Our response: we appreciate the reviewer very much for offering us such constructive suggestions. MBA is *N, N'*-methylenebis acrylamide, which was obtained from Drug Group Chemical Reagent Co., Ltd., China.

14. In the release experiment, what is the concentration of total ions? Is PBS the buffer used.

Our response: we appreciate the reviewer very much for offering us such constructive suggestions. PBS the buffer was used. However, we are sorry that due to the limitations of the current experimental conditions and time, we cannot test the concentration of total ions. We will focus on this method suggested by reviewers in our later work. We appreciate the reviewer very much again for offering us the important and constructive suggestions. The following references were referred to when preparing the release media

[29 in manuscript] Wang Q, Xie XL, Zhang XW. 2010. Preparation and swelling properties of pH-sensitive composite hydrogel beads based on chitosan-g-poly (acrylic acid)/vermiculite and sodium alginate for diclofenac controlled release. *Int. J. Biol. Macromol.* 46: 356-362.

[41 in manuscript] Wang Q, Zhang JP, Wang AQ. 2009. Preparation and characterization of a novel pH-sensitive chitosan-g-poly (acrylic acid)/attapulgit/sodium alginate composite hydrogel bead for

controlled release of diclofenac sodium. Carbohydr. Polym. 78: 731-737.

We appreciate for Reviewers' warm work earnestly and hope that the correction will meet with approval. Once again, thank you very much for your comments and suggestions.

Reviewer: 2

1. What are the error bars in Figure 7, 8, 9?

Our response: Thank you very much for reviewers' constructive suggestions. To assess reproducibility of this analysis, swelling/release experiments were replicated in triplicate for hydrogel samples. Reviewers' constructive suggestion, the precision of the three replicated experiments was determined by error bars finally.

Fig. 7. Variation of swelling ratio for CMS-g-PAA/10% PGS /ST/SA composite hydrogel beads at different pH solution. Weight ratio of each component in the CMS-g-PAA/10% PGS/ST/SA composite hydrogel beads is that CMS-g-PAA/10% PGS : SA : ST is 1 : 2 : 5; CaCl₂ concentration is 5 wt%.

Fig. 8. The cumulative release of BH from CMS-g-PAA/PGS/ST/SA composite hydrogel beads with different PGS content at 37 ± 0.5 °C under 100 rpm after 12 h. Weight ratio of each component in the

CMS-g-PAA/PGS/ST/SA composite hydrogel beads is that CMS-g-PAA/PGS : SA : ST is 1 : 2 : 5; CaCl₂ concentration is 5 wt%.

Fig. 9. *In vitro* cumulative release of BH from CMS-g-PAA/10% PGS/ST/SA composite hydrogel beads at various pHs at 37 ± 0.5 °C under 100 rpm after 12 h. Weight ratio of each component in the CMS-g-PAA/10% PGS/ST/SA composite hydrogel beads is that CMS-g-PAA/10% PGS : SA : ST is 1 : 2 : 5; CaCl₂ concentration is 5 wt%.

2. Do you have statistical analysis of Figure 2 and Figure 4?

Our response: Thank you very much for reviewers' constructive suggestions. The Figure 2 and Figure 4 do not have statistical analysis due to belonging to the characterization of the material. In

addition, as far as we know, this has not been reported in a large amount of previous literature. The literatures related to Figure 2 and Figure 4 are as follows:

[1] Cheng S, Liu XM, Zhen JH, Lei ZQ. 2019. Preparation of superabsorbent resin with fast water absorption rate based on hydroxymethyl cellulose sodium and its application. *Carbohydr. Polym.* <https://doi.org/10.1016/j.carbpol.2019.115214>.

[2] Kenawy E, Seggiani M, Cinelli P, Elnaby HMH, Azaam MM. 2020. Swelling capacity of sugarcane bagasse-g-poly(acrylamide)/attapulgit superabsorbent composites and their application as slow release fertilizer. *Eur. Polym. J.* <https://doi.org/10.1016/j.eurpolymj.2020.109769>.

[3] Behrouzi M, Moghadam PN. 2018. Synthesis of a new superabsorbent copolymer based on acrylic acid grafted onto carboxymethyl tragacanth. *Carbohydr. Polym.* 202: 227-235.

3. I don't see much difference in Figure 6. Could you explain more on Figure 6?

Our response: Thank you very much for reviewers' constructive suggestions. The change of the curves in Figure 6 is similar, but clay content has very effect on the swelling performance of the CMS-g-PAA/PGS/ST/SA hydrogel beads. As shown in Fig. 6, when 10 wt% of PGS was introduced in the CMS-g-PAA/PGS composite material, the swelling ratio of the CMS-g-PAA/10 wt% PGS/ST/SA hydrogel beads reached maximum. However, the swelling ratio of the CMS-g-PAA/40 wt% PGS/ST/SA hydrogel beads was the lowest among the hydrogel beads from 10 wt% to 40 wt%. Compared with CMS-g-PAA/ST/SA hydrogel beads, the equilibrium swelling rate of CMS-g-PAA/10 wt% PGS/ST/SA hydrogel beads significantly increased because moderate PGS was able to impair the hydrogen-bonding interaction and the entanglement of the polymeric chains, which improved the swelling ratio of hydrogel beads. Nevertheless, excess PGS took up a position of crosslinking points in hydrogel beads by large amounts of -OH groups on the surface of PGS. Meanwhile, the excessive PGS may enhance the crosslinking density of composite and plugging network voids and minimize the swelling ratio.

4. There are several issues with the grammar and English. Please revise the manuscript thoroughly.

Our response: Thank you very much for reviewers' constructive suggestions. The issues with the grammar and English were revised in manuscript thoroughly and marked in blue.

5. A large body of latest literature on biopolymer-based materials for biomedical applications is missing and must be discussed and referenced in their work. This is important so that readers are aware of the current literature. They are:

- 1) Treatment of Neurodegenerative Disorders through the Blood-brain Barrier using Nanocarriers. *Nanoscale* 2018, 10(36): 16962-16983.
- 2) Alternative Foaming Agents for Topical Treatment of Ulcerative Colitis. *Journal of Biomedical Materials Research Part A* 2018, 106(5): 1448-1456.
- 3) Non-genetic Engineering of Cells for Drug Delivery and Cell-based Therapy. *Advanced Drug Delivery Reviews* 2015, 91(1): 125-140.
- 4) Intestinal Organoids Containing PLGA Nanoparticles for the Treatment of Inflammatory Bowel Diseases. *Journal of Biomedical Materials Research Part A* 2018, 106(4): 876-886.
- 5) A Bio-responsive 6-mercaptopurine/Doxorubicin Based “Click Chemistry” Polymeric Prodrug for Cancer Therapy. *Materials Science and Engineering: C* 2020, 108(1): 110461- 110473.
- 6) Synthesis and In Vitro Characterization of Carboxymethyl Chitosan-CBA-Doxorubicin Conjugate Nanoparticles as pH-Sensitive Drug Delivery Systems. *Journal of Biomedical Nanotechnology* 2017, 13(9): 1097-1105.
- 7) Self-Assembled Hyaluronic Acid Nanoparticles for pH-Sensitive Release of Doxorubicin: Synthesis and In Vitro Characterization. *Journal of Biomedical Nanotechnology* 2017, 13(9), 1058-1068.
- 8) A pH-Sensitive Nanosystem Based on Carboxymethyl Chitosan for Tumor-Targeted Delivery of Daunorubicin. *Journal of Biomedical Nanotechnology* 2016, 12(8): 1688-1698.

Our response: Thank you very much for reviewers' constructive suggestions. The latest literature on biopolymer-based materials for biomedical applications was added in manuscript and marked in blue.

Related literatures as follows:

[15 in manuscript] Youssef A, Dudhipala N, Majumdar S. 2020. Ciprofloxacin Loaded Nanostructured Lipid Carriers Incorporated into In-Situ Gels to Improve Management of Bacterial Endophthalmitis. *Pharmaceutics*. doi:10.3390/pharmaceutics12060572.

[16 in manuscript] Massoumi B, Abbasian M, Jahanban-Esfahlan R, Motamedi S, Samadian H, Rezaei A, Derakhshankhah H, Farnudiyani-Habibi A, Jaymand M. PEGylated hollow pH-responsive polymeric nanocapsules for controlled drug delivery. *Polym. Int.* doi: 10.1002/pi.5987.

We appreciate for Reviewers' warm work earnestly and hope that the correction will meet with approval. Once again, thank you very much for your comments and suggestions.